# Relationship between Femur Mineral Content and Local Muscle Strength and Mass

**DOI:** 10.3390/jfmk9020069

**Published:** 2024-04-09

**Authors:** Bruno V. R. Ramos, Danilo A. Massini, Tiago A. F. Almeida, Eliane A. Castro, Mário C. Espada, Cátia C. Ferreira, Ricardo A. M. Robalo, Anderson G. Macedo, Dalton M. Pessôa Filho

**Affiliations:** 1Graduate Programme in Human Development and Technology, São Paulo State University (UNESP), Rio Claro 13506-900, Brazil; bruno.vital@unesp.br (B.V.R.R.); dmassini@hotmail.com (D.A.M.); tiagofalmeida.w@gmail.com (T.A.F.A.); elianeaparecidacastro@gmail.com (E.A.C.); andersongmacedo@yahoo.com.br (A.G.M.); 2Department of Physical Education, School of Sciences (FC), São Paulo State University (UNESP), Bauru 17033-360, Brazil; 3Laboratory of Exercise Physiology Research Group (LFE—Research Group), Universidad Politécnica de Madrid (UPM), 28040 Madrid, Spain; 4Instituto Politécnico de Setúbal, Escola Superior de Educação, 2914-504 Setúbal, Portugal; mario.espada@ese.ips.pt (M.C.E.); catia.ferreira@ese.ips.pt (C.C.F.); ricardo.robalo@ese.ips.pt (R.A.M.R.); 5Sport Physical Activity and Health Research & INnovation CenTer (SPRINT), 2040-413 Rio Maior, Portugal; 6Centre for the Study of Human Performance (CIPER), Faculdade de Motricidade Humana, Universidade de Lisboa, 1499-002 Lisbon, Portugal; 7Comprehensive Health Research Centre (CHRC), Universidade de Évora, 7004-516 Évora, Portugal; 8Life Quality Research Centre (CIEQV-Leiria), 2040-413 Rio Maior, Portugal; 9Training Optimization and Sports Performance Research Group (GOERD), Faculty of Sport Science, University of Extremadura, 10005 Cáceres, Spain; 10Faculdade de Motricidade Humana, Universidade de Lisboa, 1499-002 Lisbon, Portugal; 11Pos-Graduation Program in Rehabilitation, Institute of Motricity Sciences, Federal University of Alfenas (UNIFAL), Alfenas 37133-840, Brazil

**Keywords:** body composition, muscle strength, absorptiometry, bone density, young adults, femur

## Abstract

Among the stimuli able to prevent early decreases in bone mineralization, exercise has a noticeable role per se as the source of mechanical stimulus or through lean tissue enlargement by its increasing of tensional stimulus. However, prevention strategies, including exercise, generally do not establish the moment in life when attention should begin to be paid to bone integrity, according to age group- and sex-related differences. Thus, this study analyzed the relationship between variables from the diagnosis of total and regional body composition, muscle strength, and bone mineral content (BMC) of femurs in young adult males. Thirty-four young Caucasian men (24.9 ± 8.6 years) had their body composition and bone density assessed by dual X-ray absorptiometry. The subjects performed a one-repetition maximum test (1-RM) in a bench press, front pulley, seated-row, push press, arm curl, triceps pulley, leg flexion, leg extension, and 45° leg press for the assessment of muscle strength in upper and lower limbs in single- and multi-joint exercises. Lean tissue mass in the trunk and upper and lower limbs were related to femoral BMC (Pearson coefficient ranging from 0.55 to 0.72, *p* < 0.01), and 1-RM values for different exercises involving both upper and lower limbs also correlated with femoral BMC (Pearson coefficients ranging from 0.34 to 0.46, *p* < 0.05). Taken together, these correlations suggest that muscle mass and strength are positively linked with the magnitude of femoral mass in men, even in early adulthood. Hence, the importance of an enhanced muscle mass and strength to the health of femoral bones in young adults was highlighted.

## 1. Introduction

The body’s soft tissue components, such as lean tissue mass (LM) and fat tissue mass (FM), have been associated with relevant factors in the prevention of bone metabolism disorders, including osteopenia and osteoporosis [1,2,3]. Lean mass is responsible for mechanical stimuli capable of promoting bone mineralization (i.e., mechanical factors regulating osteogenic activity), whereas the indirect influence of FM appears through hormone modulation such as oestrogen, leptin, and insulin, which stimulate bone mineral waste (i.e., humoral factors regulating osteoblast activity) [4,5,6]. Past studies have demonstrated this association. In twins of different sexes, organized by age (45 pairs < 50 years and 48 pairs > 50 years), Makovey et al. [7] observed an explanatory potential of 52% of variance between LM and bone mineral content (BMC) and of 20% between FM and BMC [7]. Also, Lee et al. [8] reported LM as an independent variable with influence on whole-body bone mass density (BMD) as well as specific skeletal sites, which was a supposition from the associations observed between appendicular muscle mass and the whole body, pelvic region, and forearm values of BMD [8]. Nevertheless, the importance of the regional distribution of LM had already been demonstrated by a study carried out by Taaffe et al. [9] with elderly people (70 to 79 years old) from both sexes and different races [9]. It was concluded that maintaining or increasing LM is a good strategy to preserve BMD among elderly people, since it can reflect the effects of activity levels upon the skeleton, regardless of sex and race. For men in particular, these authors were able to verify that variations in LM affect the BMD of the body and the supporting parts of the body (e.g., hip, and lower limbs), but variations in regional LM only affect the parts which do not have supporting functions (e.g., upper limbs) [9].

These results are in line with Frost’s arguments [10], who defends the “mechanostat” principle, which follows Utah’s paradigm for skeletal physiology, relating mechanical stress to bone remodeling, within a normal limit of bone mass/density, which has been used to justify the cause–effect relationship between muscle strength or mass and body BMC/BMD, or that of a certain part of the body [10]. However, Utah’s paradigm also establishes that there is a set point beyond which mechanical stimulation does not lead to bone mass gains, and this has been used to justify the lack of such relationships between young healthy individuals and athletes. Nonetheless, the type of stimulus on the bone system provided by a certain sport (impact or mechanical tension) is capable of distinguishing BMC and BMD variations between healthy bone frames, as demonstrated in a review by Guadalupe-Grau et al. [11]. These authors collected information concerning young people and athletes who practiced different sports and observed that the stimulus of each sport influenced body, regional, and femoral BMC/BMD, which were different between athletes and their sedentary counterparts [11].

It should be emphasized that regular sports practice is a growing phenomenon [12] and correlations between body composition and physical fitness have been shown in sport [13], which is extremely useful information for coaches and plays a fundamental role in designing training sessions [14]. Another relevant issue is sports injuries, which are associated with complexity [15]; consequently, a holistic approach seems to be essential to increase the probability of success in this matter [16]. For example, the involvement of boys and young adults in specific sport training regimens, such as soccer, without an appropriate monitoring of the healthy progression of players’ conditioning level has increased the occurrence of injuries, most frequently compromising soft tissues, with the possibility to exacerbate impairments on both physical and technical levels [17]. In this sense, regular training monitoring is vital, and the collaborative work developed by professionals from different areas such as sports sciences, physiotherapy, and biomedical technology may represent an important support for the detection and prevention of injuries and, at the same time, improve sports performance without compromising tissue integrity [18].

Aligned with this assumption is that sport and physical activity practice during growth is associated with bone strength in adulthood even when such a practice is not maintained [19], probably due to the osteogenic effect from the mechanical stimuli, which stimulates BMC and BMD to increase in different anatomical sites, mainly through the impact on ground and muscle tension, with this last stimulus enhanced by the increase in lean body mass and strength [20]. Indeed, strength and muscle mass have been considered as mechanical stimuli with a dominant effect on BMC and BMD alterations, and have been related to other non-mechanical stimuli (metabolic or hormonal), which are able to modulate both bone and muscle metabolisms, as well as the responsiveness of bone tissue to mechanical stress, especially among young people and adults of both sexes [2,21,22,23].

These contexts confirm the importance of the type of exercise on bone mineral integrity variables (BMC and BMD), but the role of muscle strength is unclear since the specificity of exercise enhancing strength (e.g., the involvement of a single muscle or muscle groups) and the mode of stimuli (e.g., targeting the bone directly through tension or indirectly through mass loading) with a high effect on bone mineral health variables still remains to addressed. Hence, if the relationship between muscle strength and BMC is dictated by the influence of mechanical stimulus upon bone tissue, we can assume that among young individuals with normal bone health, physical strength performs a specific role in distinguishing between individuals with a more or less healthy bone structure. By exploring this hypothesis on femoral mineral integrity, the current study expected to address whether the conflicting results of the effect of muscle strength on this skeletal site might be further explained when the strength of muscles in different body regions is considered for the analysis. Therefore, this study aimed to assess the relationship between muscle strength, regional distribution of LM, and proximal femoral BMC, in an attempt to contribute to the positive effect of the planning of specific intervention strategies considering the use of tensional stimulus during resistance exercise to promote femoral mineralization health.

## 2. Method

The sample consisted of 34 men, aged 24.9 ± 8.6 years, with a height of 175.2 ± 5.1 cm and a weight of 71.5 ± 12.9 kg. The participants were selected over a three-year period through a social resistance training project. The selection criteria were male, aged between 18 and 35 years, Caucasian, and active (i.e., an involvement with regular exercise practice twice a week, at least) but with no experience in resistance training and not overweight (body mass index < 25). The selected subjects were informed about the procedures they would follow in this research and signed a consent form, authorizing their participation. This research was conducted according to international ethical standards for research in sport and exercise sciences [24] and in accordance with the Declaration of Helsinki. This study was approved by the University’s Local Ethics Committee (protocol: 70076317.1.0000.5398).

### 2.1. Body Composition and Bone Mineral Measurement

The dual X-ray absorptiometry (DXA) method (Hologic^®^ model, QDR Wi^®^ Discovery, Bedford, MA, USA) was used to obtain total and regional body composition. The software (Hologic APEX^®^ 5.6) calculated the fat mass (FM) and fat-free mass (FFM, composed of LM plus BMC) components, in grams or kilograms, for upper and lower limbs on both sides of the body (e.g., right and left legs and arms), providing regional composition variables (FM, FFM, and total mass) of upper and lower limbs, the torso, and the whole body. Hip bone sites of both sides (i.e., dual hip) are considered the region of interest (ROI) for BMC analysis of the total proximal femur (TPF) structural health [25].

The equipment was calibrated according to the manufacturer’s recommendation and the process was carried out and analyzed by an expert technician in the field. The assessment procedures followed the suggestions of Nana et al. [26], i.e., in a recumbent position, with the feet close together and arms positioned next to the torso. For the dual total hip scans, both legs were rotated 25° inward, and straps were used to adjust and hold the feet in the correct position. The DXA analyses were carried out on different days of strength measurements, but lasted no longer than 45 min, and included equipment warming, calibration, participant scanning, and results reporting to the participant.

### 2.2. Strength Measurements

The one-repetition maximum (1-RM) tests were performed using the following specific exercises: (a) bench press (BP), (b) front pulley (FP), (c) leg curl (LC), (d) leg extension (LE), and (e) 45° leg press (45LP). Besides these exercises, other upper limb exercises were analyzed, two of which were multi-joint (push press (PP) and seated row (SR)) and two of which were mono-joint (arm curl (AC) and triceps pulley (TP)). All these tests were carried out after a 15 min general warm-up, which was performed similarly for all participants and included stretching, arm, hip, and leg mobility, squats, and jumping exercises. The 1-RM test protocol followed the recommendations of Massini et al. [27] and Baechle and Earle [28]: (1) a specific warm-up preceded the first test attempt. The other attempts were performed with low loads and did not lead to concentric failure; (2) the load of the initial attempts was established based on average maximum strength indices of the lower and upper limbs, according to age and weight; (3) the participants performed at least three attempts with a 3 min rest period, increasing or decreasing the initial load from 1.1 to 4.5 kg, according to the ease or difficulty of the first movement. The heaviest weight successfully lifted represented the 1-RM reference value. The load value was represented in kilograms (kg). The 1-RM test was executed three times. In the second and third attempts, the 1-RM reference value obtained in the first test was divided into five percentiles (90, 95, 100, 105, and 110%) and performed randomly with a 3 min rest among the remaining attempts. Additionally, the participants were trained under professional supervision to perform the movements using the appropriate technique, following the recommendations of the National Strength and Conditioning Association (NSCA) [28].

### 2.3. Statistical Analysis

Normality was verified by the Kolmogorov–Smirnov test. Linear regression, using the Stepwise method, modeled the relationship between observed BMC values (as dependent factors) and anthropometric variables, regional and total body composition, and muscle strength of upper and lower limbs.

Variability and dispersion measures were tested by the adjusted R-squared (AdjR^2^—Equation (1)) and standard error estimate (SEE) between the dependent and independent variables. The AdjR^2^ accounts for the degree of overestimation of the explanatory coefficient (R^2^) by considering the ratio between the number of subjects (N) and factors (k) with an influence on the relationship [29]. All statistics procedures were carried out in the Statistical Package for Social Sciences (SPSS) 26.0 program (IBM, SPSS Statistics for Windows, Armonk, NY, USA), with a significance level of *p* ≤ 0.05.
(1)$${AdjR}^{2}=1-\left[\left(1-R^{2}\right)\times \left(\frac{N-1}{N-k-1}\right)\right]$$

The sampling power for associations between dependent and independent variables was determined considering the sample size (men = 34) in G*Power software. The input parameters were: (a) the coefficient from the variance analysis (R^2^), (b) Zα = 1.96 for an index of α = 0.05, and (c) security β = 1.282 for a sample with a minimum power of 80% (β = 0.20), according to Equation (2).
(2)Z1−β=n−312Ln1+r1−r−Z1−α2

In addition to the sample power, the analysis based on magnitude inferences was applied to test the probability of a true magnitude of an effect that is substantially positive or negative, and insignificant or trivial (with a probability rate of 66 to ensure an unequivocally useful effect, i.e., probabilities of benefit >25% and probabilities of loss <0.5%). These probabilities were analyzed qualitatively, from borderline values, according to the following scales: <1% = very unlikely; 1–5% = quite unlikely; 5–25% = unlikely; 25–75% = possibly; 75–95% = likely; 95–99.5% = quite likely; and 99.50–100% = very likely. This procedure ensures that after repeating the exercise several times, the sampling distribution of z = 0.5 ln ((1 + r)/(1 − r)) will tend approximately towards normality with a variation of 1/(n − 3) [30].

## 3. Results

Among the individuals, the fat percentage was 18.2 ± 6.4%, classifying their adipose levels as normal. Body LM and FM were 55.6 ± 7.8 kg and 13.5 ± 6.9 kg, respectively. The observed TPF BMC value was 217.1 ± 46.5 g (range: 142.7–327.7 g). Whole-body BMC was 2477.8 ± 386.6 g (range: 1754.4–3380.4 g).

Table 1 presents complete regional body composition values for all participants. Pearson’s coefficients observed between TPF BMC and LM of the trunk (r = 0.60), upper limb (r = 0.55), lower limb (r = 0.72), and whole body (r = 0.67) showed a *p*-value ≤ 0.01. Among these correlations, only the LM of lower limbs proved to be an explanatory variable for the femoral BMC values (Figure 1A,B). The determination potential of LM of lower limbs showed that AdjR^2^ = 0.52 and sampling power = 99.9%, whose effect was considered to be “very likely” to be found in other populations with the same characteristics (Figure 1A).

Figure 1B depicts the agreement between the measured values of TPF BMC with those estimated from the LM of the lower limbs, evidencing a small bias of ~2.1% when comparing predicted vs. estimated values of TPF BMC. The whole-body LM potential related with TPF BMC is lower than that observed for the LM of the lower limbs. Concerning the TPF BMC association to whole-body LM, the AdjR² = 0.45 and sampling power = 99.5% evidenced that the effect is also “quite likely” to be found when the correlation in other populations is reproduced (Figure 1C). It was found that the whole-body LM variable can estimate TPF BMC with an average bias of ~1.3% when compared to the values measured of TPF BMC (Figure 1D).

Table 2 shows muscle strength values obtained from the 1-RM test. Pearson’s coefficients between muscle strength and femoral BMC are also shown in Table 2. Significant correlations (*p* ≤ 0.05) between muscle strength and TPF BMC were observed both for exercises involving regions close to the femur (hip and knee), as well as those involving parts further away from the femur (scapular and elbow).

This local effect of muscle strength on femoral mineralization showed the physical suitability of hamstring strength as an explanatory variable for TPF BMC variance (Figure 2A,B). The potential for determining 1-RM in the leg curl exercise on the variation in BMC showed AdjR² = 0.19 and a satisfactory sampling power (=80.4%), which is an effect considered “likely” to occur in other populations with the similar profiles of body composition and strength (Figure 2A).

In Figure 2B, it can also be seen that the diagnosis of leg flexion strength agrees with TPF BMC with an average difference of ~1.7%. Additionally, when analyzing the inclusion of other muscle strength variables on TPF BMC, it was verified that the leg flexion +45° leg press increases the association effect by 4% (R^2^ = 0.25; *p* ≤ 0.01), and can reach 7% when including the leg flexion +45° leg press + bench press + seated row (R^2^ = 0.28; *p* < 0.05).

## 4. Discussion

This study aimed to assess the relationship between muscle strength, regional distribution of LM, and femoral BMC. The results showed that variations in femoral BMC in men were associated with both LM (regional or total) and maximum muscle strength in exercises involving the hip region. Data showed that the development of muscle strength in exercises involving the hip region (i.e., close to the femur) is recommended for the maintenance/development of femoral mineral integrity among men, but the remote effect demonstrated by exercises involving muscle groups of the trunk and upper limb, such as the bench press and seated row, cannot be ruled out. Consequently, regional specificity could be noted between strength and LM and bone mass, which was suggested by Matsuo et al. [31] as being a context influenced by hormonal and mechanical stimuli attributed to the action of a larger muscle mass. Previously, the role of regional fat-free tissue for monitoring the muscle strength development in specific body regions [27,32] was also evidenced, and from the current results, this association was also observed at the regional level (e.g., the specificity of the exercise involving a target region), while confirming the aforementioned role of regional and whole-body LM, but now in the context of bone sites with a weightbearing function, such as the femur.

The current findings are also aligned to the results of Taaffe et al. [9], which reported LM and strength as factors with influence on the alterations on femoral and whole-body BMD. According to Taaffe et al. [9], the alterations ranging from ~3 to ~6% in the BMD of the lower and upper limbs, femur, and whole body could be related to a 7.5 kg increase in LM among men, regardless of race. The current results evidenced how positive the local effect of muscle strength on bone mass in a given body region might be, which can be seen as a tendency for young men. Aligned with the assumption that the tensional magnitude of the muscles may account for the effectivity of the stimuli on the bone morphology [8,9,10,22], this association tends to be strengthening by muscle volume and the local or systemic stimulus generated by the muscle action, since the current study observed that exercise such as the 45° leg press (i.e., action of the back of the thigh and posterior of the hip), as well as the bench press (i.e., action of the back of the trunk and posterior of the arm) and seated row (i.e., actions of the back of the trunk and back of the arm) are positively related to an enhanced femoral mass. Also noteworthy was the collective effect of these muscles (leg flexion + 45° leg press + bench press + seated row) on TPF BMC (R^2^ = 0.28; *p* < 0.01).

Despite the current results not analyzing the possible influence of FM on bone mass, this association is well documented, showing that body fat is associated with BMC, since the secretion of androgen hormones is inversely related to the fat percentage in the lower limbs, and also that android body fat distribution (which is more common in, but not limited to men) promotes some stimulation of the bone structures in the trunk [31]. However, the FM effect on the bone might not be a confounding factor between the sample of subjects in the current study, since no body fat excess was observed between the participants. In addition, the effect on bone mass has been better explained by the mechanical–metabolic–hormonal triad, both for local and remote tension stimulation, by the interaction between muscle activity and the secretion of sex hormones (testosterone and oestrogen) [5,11,21,22,33] and between muscle activity and the secretion of sex hormones and the secretion of anabolic hormones (insulin and growth hormone), which modulate bone and muscle metabolism, or simply modulate the responsiveness of bone tissue to mechanical stress, mainly in young people and adults of both sexes [2,22,23,34,35]. This aforementioned assumption reinforces the inference from the current findings that muscle strength and mass exert an important role on bone mass integrity.

Indeed, this effect has been demonstrated experimentally for local stimuli, i.e., muscle mass loss preceding bone mass loss, when it falls into disuse, as well as the recovery of muscle mass prior to normalization of bone mass and density, as observed by Sievanen et al. [36] in a study involving rehabilitation of the strength in knee muscle extensors and the concomitant response of BMD in patellar bone variation after injury to the posterior cruciate ligament in a physically active young woman. Thus, this study proved the cause–effect relation, directly and indirectly from other body factors, which supports the idea of the mechanical effect as modulating the activation of the system of bone turnover, adjusting the balance for bone mass gains [21]. This is particularly interesting since conflicting reports suggested that resistance training had neither a positive nor a negative effect on bone mineral integrity in a healthy older population of both sexes [37].

Aligned to the role of muscle strength on BMC, Matsui et al. [38] concluded the need to develop quadricep strength to prevent loss of femoral neck bone mass in men (n = 763) aged from 40 to 81 years, recommending both specific exercises for the region (due to direct action of mechanical tension on the bone) and general physical activity (due to the systemic action of physical activity on bone metabolism). Finally, Guimarães et al. [39] also report local and remote associations between the strength of multi-joint exercises and BMD in male university students (~25 years). According to these authors, the strength required for the action of pushing (chest press) and pulling (high rower), involving the large muscles of the upper limb, is not only a good predictor of upper limb BMD, but also of lower and total BMD. These authors have also demonstrated that pushing strength involving lower limb muscles (45° leg press) also presents a local (lower limb BMD) and a remote effect (upper limb and total BMD), but with less potential compared to upper limb actions. Thus, these authors concluded that muscle mass and strength are components of physical fitness which must be developed and maintained by repeating resistance exercises in order to preserve a healthy BMC and BMD throughout life. Complementary to this, the current findings add the information that femoral mass integrity is also influenced by the strength and lean mass of the lower-limb region.

Indeed, the resistance training performed by young men (20–29 years) and planned with upper and lower limb exercises, for six months and with loads ranging from 12 to 15 RM, was able to significantly increase the BMD of the femur and other parts of the body, which seems to be linked with the increasing LM and muscle strength in each exercise [40]. In addition, in middle-aged men (54–61 years), Huuskonen et al. [41] also found a 3.8% increase in femoral BMD, after four months of planned resistance exercises, practiced three days a week and with a load between 5 and 15 RM [41]. Collectively, these studies confirm the beneficial nature of strength training in regulating BMC/BMD. In general, resistance exercises tend to promote improvements in bone mass in local and remote regions in relation to the region that is affected in the exercise, with a recommended intensity of between 70 and 90% 1-RM (2–3 sets per exercise, 1–3 min of rest time between exercises, 2–3 sessions per week) [11]. In addition, if possible, strength exercises should be combined with other impact exercises (such as jumping, and combat sports) but avoid long-duration aerobic exercises (endurance) and exercises in a low-gravity environment (such as swimming), since these tend to decrease osteogenic activity [11].

Limitations of this study include the small sample size and the lack of women and participants with a higher strength conditioning level. Therefore, it would be interesting to consider whether regular strength training could influence the relationships found between strength and muscle mass and proximal femoral BMC. In addition, the lack of information about the history of sports activities performed in previous periods of life is another limitation in the current study, considering that it would allow the quantification of the level of influence of the current and past levels of physical activity or exercise involvement on the measures of regional and whole-body BMC [19]. Despite these limitations, based on the present findings, it is recommended to choose exercises aiming to stimulate the action of the posterior and anterior muscles of the thigh, the posterior muscles of the hip, the posterior and anterior muscles of the trunk, and the muscles of the arms when preparing a physical activity for young adults who are not involved with resistance training. Therefore, not only resistance exercise but also different sporting stimuli have an influence on regional and total BMC. The mechanical factor, which stems from strength and muscle mass, is therefore accepted as the predominant stimulus for bone tissue stability and resistance, according to Frost [10], Beck et al. [4], and Schoenau [34]. Future studies should be able to better explain the role of each exercise in randomized populations of different races, ages, and strength fitness levels, as well as evaluating the metabolic profile of bone tissue by looking for interactions between mechanical stress and bone metabolism.

## 5. Conclusions

In young adult males, maximal muscle strength, analyzed in different resistance exercises involving a wide range of joint actions and muscle recruitment, was found to be associated with variations in femur bone mass. This effect illustrates the importance of both the direct stimulus (local specification) and the remote stimulus (systemic stimulus resulting from intense exercise). In this context, the importance of muscle mass and strength in the mineral integrity of bone tissue was defined, as well as the role of activation of large muscle groups during exercise, both of which are able to contribute at local or systemic levels with the enhancement of the femoral health integrity in young men.

## Figures and Tables

**Figure 1 jfmk-09-00069-f001:**
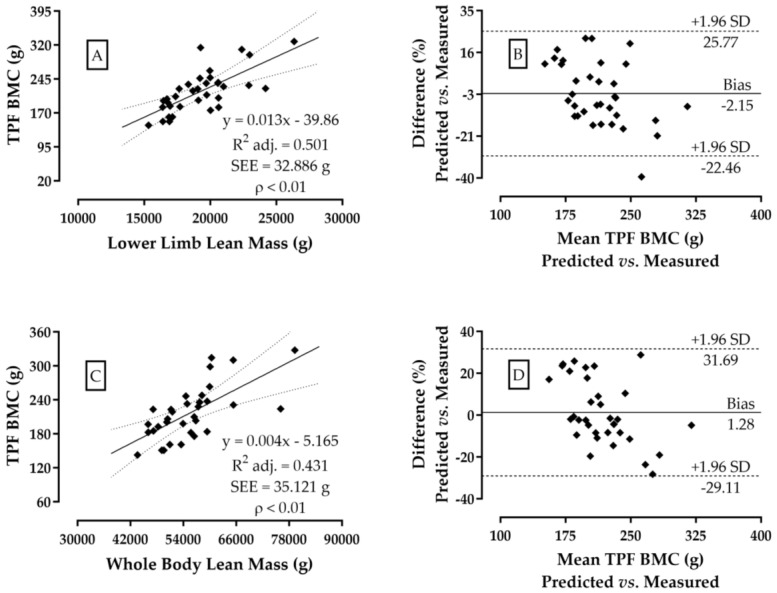
Dispersion and agreement analysis (Bland–Altman) between TPF BMC and the regional lean mass of lower limbs (**A**,**B**) and whole-body lean mass (**C**,**D**), in men. N = 34. (Note: BMC: bone mineral content).

**Figure 2 jfmk-09-00069-f002:**
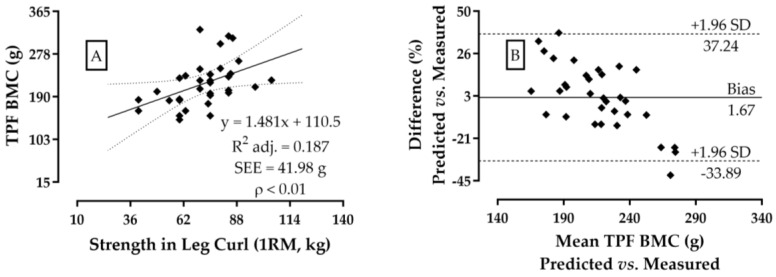
Dispersal and agreement analysis (Bland–Altman) between muscle strength in the leg flexion and TPF BMC (**A**,**B**), in men. N = 34.

**Table 1 jfmk-09-00069-t001:** Average values ± SD (minimum and maximum) for regional composition. N = 34.

	Total Mass (kg)	Lean Mass (LM, kg)	Fat Mass (FM, kg)
Trunk	32.7 ± 7.2 (23.0–59.9)	26.0 ± 4.2 (19.2–39.6)	6.0 ± 3.8 (2.0–19.3)
* Upper limb	8.6 ± 1.7 (6.3–14.4)	6.8 ± 1.2 (5.3–11.6)	1.4 ± 0.83 (0.45–4.7)
* Lower limb	25.2 ± 4.2 (19.0–39.4)	19.2 ± 2.5 (15.3–26.3)	5.1 ± 2.5 (1.8–11.8)

* Values of mass considered the measurements in both legs and arms for lower and upper limbs, respectively. SD: standard deviation.

**Table 2 jfmk-09-00069-t002:** Average values ± SD (minimum and maximum) for 1-RM strength (kg) and Pearson’s coefficients between muscle strength values and femoral BMC. N = 34.

Exercises	Average ± SD	Minimum–Maximum	TPF BMC (g)
Bench Press (kg)	60.0 ± 16.0	40.0–112.5	0.34 *
Front Pulley (kg)	62.2 ± 16.4	40.0–130.0	ns ^‡^
Leg Curl (kg)	72.0 ± 14.5	40.0–105.0	0.46 **
Leg Extension (kg)	91.8 ± 30.3	40.0–188.0	ns ^‡^
45° Leg Press (kg)	259.1 ± 57.5	120.0–366.0	0.39 *
Push Press (kg)	40.1 ± 13.0	18.0–82.0	0.38 *
Seated Row (kg)	71.8 ± 21.7	35.0–150.0	0.39 *
Triceps Pulley (kg)	54.4 ± 13.9	35.0–100.0	0.36 *
Arm Curl (kg)	35.1 ± 8.8	16.0–65.8	ns ^‡^

Observation: significant correlation is * *p* ≤ 0.05 or ** *p* ≤ 0.01. ^‡^ The acronym “ns” shows no significant correlation for stipulated levels. BMC: bone mineral content; SD: standard deviation.

## Data Availability

The data that support the findings of this study are available from the corresponding and last author (mario.espada@ese.ips.pt and dalton.pessoa-filho@unesp.br) upon reasonable request.

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
