# Peer review of "Relationship between Femur Mineral Content and Local Muscle Strength and Mass"

_jfmk, 2024, doi:10.3390/jfmk9020069_

Round 1

Reviewer 1 Report

Comments and Suggestions for Authors

Congratulations to the authors for their work. It provides information on the benefit of strength training on bone mineral density. The design seems appropriate, the instruments for measuring the dependent variables are also adequate and the correlational analysis methodology is appropriate. The presentation of the results is clear and the conclusions are supported by the results.

Author Response

I and my fellow authors would like to thank you for reviewing this manuscript, and accept it for publication.

Reviewer 2 Report

Comments and Suggestions for Authors

General comment
The Authors analysed the relationship between femoral BMC, local muscle mass and strength. The topic is of high importance since new information on this relationship could help us to understand the main risk factors of not normal skeleto-muscular development of the human body and the injuries of this organ system. The aims are well defined. I suggest the manuscript for publication after a major revision. The main reasons of my suggestion are:
1) One of the limitations of the presented study that sample was too small for an analysis planned in the objective of the paper. It is not possible to correct this limitations, I know, but the results of the analysis should be more carefully interpreted.
2) It should be justified why the relationship between the strength of the upper extremities and the femoral BMC was analysed, without an explanation of this analysis it seems unnecessary.
3) BMC and BMD indicators of bone development are used in the manuscript, it seems in the Discussion section as if they would be the same indicators, it is suggested to clear the difference between them in the Introduction or Methods section.

I listed my comments and suggestion by the order of the sections of the manuscript.

Abstract
A1: “Significant correlations between 1-RM values for different exercises and femoral BMC (R² between 0.34 and 0.46, p<0.05), suggests that muscle strength and mass in body regions surrounding or remote from the hip effectively stimulates femoral mass in men, even in early adulthood” – it is not clear why not only the relationship is mentioned in the Abstract, in this case more robust femoral (BMC) and bigger muscle mass development could result in bigger muscular strength.

Introduction
I1: “Recently it was stressed that non-contact injuries occur more frequently in male youth soccer players, with the knee and ankle indicated as having the highest incidence.” – It is not clear, it is stated that more frequently but not mentioned compared to whom?

Methods
M1: “in a recumbent position, with the feet close together and positioned next to the torso.” – feet were positioned next to the torso? Please correct the sentence.
M2: “The interventions were carried out on different days but were completed within a week and lasted no longer than 45 minutes.” – what kind of interventions? Why did the DEXA examinations take so long?
M3: “Besides these exercises, other upper limb exercises were analyzed, two of which were multi-joint (push press (PP) and seated row (SR)) and two mono-joint (arm curl (AC) and triceps pulley (TP)).” – why these measurements are important to mention them? Only the lower extremity is studied in the paper.
M4: “Variability and dispersion measures were tested by the sample-adjusted coefficient (AjR2 – Eq. 1) and standard error estimate (SEE) between the dependent and independent variables.” – please justify why the adjustment was important to use in this small sample.
M5: The BMC in the femur could be estimated by the DEXA method or BMC estimation was possible only for the whole lower extremities?
M6: The level of physical activity in the studied group should be described.

Results
R1: Table 1: data are valid for only one upper and only one lower limb or for both of them?
R2: “Pearson’s coefficients observed between BMC and LM of the trunk (r = 0.60), upper limb (r = 0.55), lower limb (r = 0.72), and whole-body (r = 0.67) showed a p-value ≤ 0.01.” – see comment R1
R3: “Figure 1 (panel B) also analyzes the correspondence between the estimate of femoral BMC and LM of the lower limbs, which shows an average difference of 2.1% when compared to the real values.” – this sentence is meaningless, please correct it (figure does not analyze, shows the average difference? what kind of difference?)
R4: “Concerning femoral BMD variation, whole-body LM showed …” – why BMD? BMC was used before.
R5: “It was found that whole-body LM was able to explain the diagnosis of femoral BMC with an average difference of 1.3%” – again, a meaningless sentence (diagnosis? what kind of difference?), please correct it.
R6: It is not clear whether the measurements were taken on only one extremity or both, on the right and the left sides, and if only one extremity was measured, which one (dominant or always the left or the right side?), please complete the Methods section and the Results section with this information.

Discussion
D1: “… to the insertion torque of its three muscles directly on the ischium bone.” – os ischium exists only in children before the ossification of the 3 parts of os coxae, the study was carried out in adults, so this sentence should be corrected.
D2: “it seems to be more related to muscle volume and the local or systemic stimulus generated by the muscle action, since exercise such as the 45º leg press (i.e., action of the back of the thigh and posterior of the hip), as well as bench press (i.e., action of the back of the trunk and posterior of the arm) and seated row (actions of the back of the trunk and back of the arm) also proved to be effective.” – this statement is not convincing, better results in strength measurements are not direct evidence for their effectivity on the BMC development in the region, only the relationship was confirmed in a very small sample. This statement should be corrected.
D3: “Indications are showing that not only mass distribution, but also body fat is associated with BMC, since the secretion of androgen hormones is inversely related to the fat percentage in the lower limbs, indicating that android standard of body fat distribution (which is more common, but not limited to men), promotes some stimulation of the bone framework in the trunk region [23]. But the effect on bone mass has been better explained by the mechanical-metabolic-hormonal triad, both for local and remote tension stimulation, by the interaction between muscle activity and the secretion of sex hormones (testosterone and estrogen) [5,11,25,26,27] and between the last two and the secretion of anabolic hormones (insulin and growth hormone), which modulate bone and muscle metabolism, or simply modulate the responsiveness of bone tissue to mechanical stress, mainly in young people and adult of both sexes [2,27,28,29,30].” – this paragraph should be deleted, the manuscript does not report any results on analyses of the relationship between fat mass and bone development.

Author Response

I and my fellow authors would like to thank you for reviewing this manuscript. Please, see the responses to your comments in the file attached.

Reviewer 3 Report

Comments and Suggestions for Authors

Despite the fact that the title of the manuscript promises to examine the relationship between the mineral content of the femur and the associated muscle mass-muscle strength, other irrelevant problems are also discussed (upper limb, fat mass).

The topic of research is interesting, in general, the effect of strength development on the skeletal system is current and of interest.  Unfortunately, the authors were only partially able to solve the task set before their own seeds, the work needs correction.

 It is necessary to agree with the hypothesis that muscle work affects the characteristics of the skeletal system, however, the only information about the training history of the selected sample is that they have been doing strength development work for at least 3 years. It is difficult to estimate the effect based on this.  Known, that the somatotype, the body composition is under a strong genetic influence.

The measurement methods used are correct. Does it confuse body composition data as to what effect of intervention (strength development) should be assumed, bone structure, bone mass and mineral content? There are several irrelevant statements in the manuscript, such as the discussion of measured and regression estimated variables, which is otherwise unnecessary analysis based on correlation values. The limitations of the work are considerable, as the authors describe. I see the main reason for this in the fact that the discussion of the topic and the measured results is not limited to the research provided, they could not examine the effect of training.  The discussion should primarily include the relationship between muscle mass, muscle strength and femoral properties. It is necessary to take into account the tasks of the femoral that are related to the support system. The topic of the thesis is promising, but the authors could not take advantage of it, the thesis needs correction.

Author Response

(The authors gave the same response as above.)

Reviewer 4 Report

Comments and Suggestions for Authors

I would like to appreciate the efforts of the authors in implementing the project and writing this article „Relationship Between Femur Mineral Content and Local Muscle Strength and Mass“.

This study aimed to assess the relationship between muscle strength, regional distribution of lean tissue mass, and femoral  bone mineral content, in an attempt to contribute to the planning of specific intervention strategies considering the use of tensional stimulus of the resistance exercise to ensure femoral mineralization health.

This article provides interesting information useful for practice. However, some details need to be explained.

Introduction: The information given is fine, but you need to put it behind you, relate to the topic of the work. I recommend improving the axis of thought. The information needs to be better linked and relate to the topic.

Method: Why were 34 participants included? Was the power analysis done? Sample size estimation?

Line 106: Can you describe or quote the selection of participants through a social resistance training project?

Line 109: Can you define obesity? What level of obesity?

Line 109: Missing a period after a sentence.

Measurement:

Line 117: Extra bracket.

Line 117: Is it not clearly specified BMC of what exactly, whether whole body or femoral? It must be clearly written here.

Line 122: I recommend a quote.

Line 124-126: This must be more itemised and clearly specified. The intervention is not sufficiently specified, it is important for the article.

Strength Measurements:

Line 129: 1-RM - the explanation of the abbreviation is only in the abstract, not in the text.

Line 133: It is necessary to define or quote a warm-up. Is it 15 minutes of breakdown? Or 15 minutes of mobility training? Or a combination? And did all participants have the same warm-up?

Line 135: "The others" is meant during warm-ups or exercises?

Line 145: How were they trained? Who taught them? How did they solve individual differences in the style of performance?

Results:

Line 183: Why was BMC evaluated at work and not BMD? BMC is just content and does not take into account the size of a given segment, so the information is not as telling about the state of the bones.

Line 186-187: It might be more appropriate to include correlation values in the table.

Line 193: Somewhere the space before and after =, somewhere without spaces, I recommend unifying.

Line 198: That would be more of a discussion.

Line 199: BMD is listed here, there is no mention of it in the methodology.

Line 242: Was different height of the subjects taken into account? Or doesn't it matter?

Lines 249-250: What do they mean "same characteristic".

Author Response

(The authors gave the same response as above.)

Round 2

Reviewer 3 Report

Comments and Suggestions for Authors

Thank you, I accept all corrections.

Reviewer 4 Report

Comments and Suggestions for Authors

All my questions have been answered, comments have been taken into account.